# The Relationship between Mobile Phone Dependence and Subjective Well-Being of College Students in China: A Moderated Mediation Model

**DOI:** 10.3390/healthcare11101388

**Published:** 2023-05-11

**Authors:** Guangming Li

**Affiliations:** 1Key Laboratory of Brain, Cognition and Education Sciences, Ministry of Education, South China Normal University, Guangzhou 510631, China; lgm2004100@m.scnu.edu.cn; 2School of Psychology, Center for Studies of Psychological Application, Guangdong Key Laboratory of Mental Health and Cognitive Science, South China Normal University, Guangzhou 510631, China

**Keywords:** college students, mobile phone dependence, subjective well-being, self-esteem, social support

## Abstract

Many studies have shown that mobile phone dependence decreases subjective well-being, but there have been relatively few studies that investigate the specific mechanisms between mobile phone dependence and subjective well-being. In this study, the mediating effect of self-esteem and the moderating effect of social support were investigated to explore the specific mechanisms between mobile phone dependence and subjective well-being. The objective of the study is to explore the mechanism of mobile dependence on subjective well-being by constructing a moderated mediation model. College students from twenty classes in three universities were randomly selected. A total of 550 college students fully participated in the actual evaluation and completed the general well-being scale, mobile phone addiction index scale, self-esteem scale, and social support scale. SPSS17.0 was used to analyze the data. The results show that: (1) Self-esteem partially mediates the relationship between mobile phone dependence and subjective well-being. Mobile phone dependence not only has a direct influence on subjective well-being, but also influences subjective well-being indirectly by self-esteem; (2) The mediating effect of self-esteem between mobile phone dependence and subjective well-being is moderated by social support. Social support moderates the second path of the mediation, and the higher the social support, the greater the degree of self-esteem on subjective well-being. For the management of mobile phone dependence of college students, more attention needs to be paid to the personality characteristics of different students. In addition, there should be efforts to avoid blindly educating students and instead to increase their social support and create a good atmosphere on campus and in society. Only in this way can they improve their subjective well-being.

## 1. Introduction

### 1.1. Background

With the advancement of technology, mobile phones have developed rapidly. Mobile phones have become an indispensable communication tool in people’s lives [1,2]. Just as everything has two sides, mobile phones will not only facilitate people’s lives, but also cause many problems [3]. Studies have confirmed that excessive use of mobile phones can lead to problems, such as sleep disorders, depression, and social behavior disorders [4,5,6,7,8].

The concept of Internet addiction was first proposed by Goldberg, sometimes known as mobile phone dependence, pathological network use, nomophobia, etc. [9,10,11,12,13,14,15,16]. Although the terms used to describe Internet addiction are different, their connotations are basically the same, mainly referring to individuals’ unrestrained Internet use, which also has a negative impact on their lives. Internet addiction has become a worldwide problem, which is attracting considerable attention of psychologists [17,18,19]. Mobile phone dependence, which is considered a form of Internet addiction, is when individuals indulge in mobile activities and cannot extricate themselves and control their own impulses to use mobile phones. Some scholars have suggested that mobile phone dependence is similar to addictive behavior, which will negatively influence the individual’s physical and mental health [20,21,22]. By referring to the relevant literature, mobile phone dependence is closely associated with subjective well-being, and those individuals with high degrees of mobile phone dependence will have lower subjective well-being [23]. At present, some studies have explored the relationship between mobile phone dependence and subjective well-being [9,17,24], but there are relatively few studies that explore the specific mechanisms of mobile phone dependence on subjective well-being. That is not enough. Based on the previous studies, the specific mechanisms of mobile phone dependence on subjective well-being and the working conditions were explored. The former is mainly about the mediating effect, and the latter is about the moderating effect. Studies of the mediating effect help to elucidate how mobile phone dependence influences subjective well-being, and studies of the moderating effect help to clarify when it works [4,25]. Studies of the moderated mediation model can reveal the above two issues simultaneously, that is, how mobile phone dependence influences subjective well-being and when it works [4]. It is possible to express nomophobia as the fear and panic experienced by individuals when they cannot reach the smart device [26]. Nomophobia occurs in individuals when their access to a mobile device is interrupted and disconnection of the communication source, deprivation of access to the internet, or anxiety to move away from social networks cause involuntary fear [27].

In recent years, with the development of positive psychology, more and more people are paying attention to their own positive experience, and many psychologists have focused on individual’s psychological state and emotional experience from the view of positive psychology. Under such a background, subjective well-being has gradually become a hot topic in psychology. Although many psychological experts were concerned about subjective well-being and its influencing factors, these studies were less concerned about the specific mechanisms between mobile phone dependence and subjective well-being. The objective of the study is to explore the mechanisms of mobile dependence on subjective well-being by constructing a moderated mediation model.

### 1.2. Model

Ecological systems theory believes that individuals are in a multilevel social system, which means that the development of the individuals is not only influenced by one’s own characteristics (e.g., personality, such as self-esteem), but also by the external environment (e.g., social culture) [28]. In other words, the development of the individual is influenced by the interaction between humans and the environment. When the individual can obtain support from the ecosystem, the ecosystem will promote the development of the individual; otherwise, it will hinder the individual’s development. This study used the concept of ecological systems theory, which states that the development of the individual is influenced by the interaction between humans and the environment, to explore the influence of humans and the environment on subjective well-being.

In this model, the “culture” only refers to social culture. However, according to ecological systems theory [28], one individual is embedded within four nested systems, the mircosystem, mesosystem, exosystem, and macrosystem, which are distinguished by their distances from the individual and the extent they may exert a direct influence on an individual. When it comes to the influences that the environment exerts, researchers often mistakenly only think of the “big picture”. For example, in some cross-cultural studies, researchers only considered the influence that different social cultures had, such as individualism and collectivism. However, microsystems are the contexts in which people have primary face-to-face contact with important and impactful individuals. Social cultures may be the most stable and profound in microsystems, and it is also the carrier of culture [29]. So, it is reasonable to say that the social culture plays a significant role in the development of people’s psychology (mobile phone dependence and subjective well-being) and personality (self-esteem). Therefore, it is necessary to contain social factors when exploring the relationship between subjective well-being and personality.

In terms of the individual level, the impact of self-esteem is particularly pronounced in many studies, which examined the influence of individual factors on subjective well-being [30,31,32,33]. Self-esteem could be used as the assessment of self-worth [34]. According to the relevant literature, there was a positive correlation between self-esteem and subjective well-being, and those individuals with high subjective well-being might also have high self-esteem [35,36]. In addition, mobile phone dependence was negatively associated with self-esteem, the higher the degree of mobile phone dependence, the lower the self-esteem [20,37,38]. There may be a significant relationship between individual self-esteem and mobile phone dependence, and self-esteem had a significant impact on subjective well-being. Thus, mobile phone dependence may reduce the individual’s self-esteem and then reduce the individual’s subjective well-being. In other words, self-esteem can be seen as a mediator between mobile phone dependence and subjective well-being.

As for the social environment, social support reflects the individual’s social development, which is an important variable influencing the development of individuals. Therefore, social support, an important variable reflecting the society and environment, is an important predictor of subjective well-being. Domestic and foreign psychologists have unanimously concluded that social support has an important effect on subjective well-being [39,40,41]. Although social support may influence subjective well-being, it is impossible that all adolescents are equally influenced. Studies have confirmed that there are also differences in subjective well-being among individuals with different social support [42,43]. In addition, mobile phone dependence will affect the individual’s self-esteem, but social support is conducive to improving the individual’s self-esteem, to have a more complex impact on subjective well-being. According to the above description, the framework of this study is shown in Figure 1.

The variable relationship in Figure 1 is based on two reasons. On the one hand, the relationship was obtained from previous research [9,17,39,40,41]. On the other hand, through competition among models from the real forecast data results obtained, Figure 1 is optimal. Therefore, both experience and practice show that Figure 1 may be more feasible or more reasonable.

### 1.3. Hypothesis

Mobile phone dependence will negatively influence the individual’s health, such as subjective well-being [20,21,22]. Mobile phone dependence was negatively associated with self-esteem, and the higher the degree of mobile phone dependence, the lower the self-esteem [20,37,38]. Those individuals with high subjective well-being might also have high self-esteem [35,36]. Mobile phone dependence may reduce the individual’s self-esteem and then reduce the individual’s subjective well-being. Social support has an important positive effect on subjective well-being [39,40,41], and there are also differences in subjective well-being among individuals with different social support [42,43].

This study assumed that self-esteem is a mediating variable and social support is a moderating variable and assumed the following hypotheses:

**Hypothesis** **1.**
*Mobile phone dependence has a significant negative impact on subjective well-being.*


**Hypothesis** **2.**
*Mobile phone dependence has a significant negative impact on self-esteem.*


**Hypothesis** **3.**
*Self-esteem has a significant positive impact on subjective well-being.*


**Hypothesis** **4.**
*Self-esteem plays a mediating role between mobile phone dependence and subjective well-being.*


**Hypothesis** **5.**
*Social support plays a moderating role between self-esteem and subjective well-being.*


This study explores the mechanism of mobile dependence on subjective well-being by constructing a moderated mediation model. This study assumed that self-esteem is a mediating variable between mobile phone dependence and subjective well-being; that is, mobile phone dependence indirectly affects subjective well-being through self-esteem. Social support is a moderating variable, which moderates the mediating effect of self-esteem between mobile phone dependence and subjective well-being.

## 2. Methods

### 2.1. Participants

The participants were from a sample of Chinese college students (*N* = 605) between 19 and 27 years of age. Of these, 550 college students (90.9%) from 20 classes in 6 different departments fully participated in the actual evaluation. All of them completed the general well-being scale, mobile phone addiction index scale, self-esteem scale, and social support scale. Among the participants, 189 were male (34.4%), and 361 were female (65.6%). The value range of age was 19–27; the average of age was 23.56, and the deviation of age was 3.44. Han ethnicity was 369 (67.09%), and Non-Han ethnicity was 181 (32.91%). Higher income was 151 (27.45%); middle income was 203 (36.91%), and low income was 196 (35.64%). A sample of 550 students was involved. Descriptive statistics of demographic measurements are shown in Table 1.

### 2.2. Procedure

The college students from twenty classes were asked to evaluate four scales that were selected based on some corresponding literature [34,44,45,46]. All these classes were selected randomly. Experimenters had been trained before the survey was administrated to students. The college students from twenty classes in three universities were randomly selected. Data were collected during the class using a paper/pencil version survey administered to all students in these school classes. Students’ physiological indicators (e.g., height, weight, and blood pressure) were assessed within the first week following the baseline measurements. The academic affairs offices of the three universities provided the measurement sites. Students were wearing light clothes and were barefoot when measured in air-conditioned rooms. Research staff was trained before they administered the survey.

Informed consent was obtained from all participants. All participants were voluntary and adequately informed of the aims, methods, sources of funding, any possible conflicts of interest, institutional affiliations of the researcher, anticipated benefits, potential risks of the study, etc. For this study, all human participants were anonymous. This study conformed to generally accepted scientific principles and was approved by the South China Normal University (SCNU) research ethics board (Institutional Review Board). The ethics board of SCNU approved the experiments including any relevant details and confirmed that the data from human participants contained in the manuscript were collected anonymously. The ethics board of SCNU confirmed that all experiments were performed in accordance with relevant guidelines and regulations.

### 2.3. Measurements

#### 2.3.1. Subjective Well-Being

In this study, general well-being schedule was used, which was developed by Fazio [47], to investigate subjective well-being. The general well-being schedule is a self-rated questionnaire, which is used to assess the individual’s subjective well-being [48]. The general well-being schedule was developed by American psychologists, and it has been widely used [49,50]. Many studies have shown that the general well-being schedule has good reliability and validity [24,48,49]. In order to adapt it to Chinese culture, Chinese scholars revised the general well-being schedule [44]. The Chinese version of the general well-being schedule consists of 18 items, including 6 factors and the higher the score, the higher the level of subjective well-being. It is worth noting that there are several reverse-scoring items in the schedule, so they should be reversed in the specific calculation. In China, the revised general well-being schedule has been widely used and has shown great reliability [24,44,51]. In this study, the Cronbach’s alpha of the general well-being schedule was 0.84.

#### 2.3.2. Mobile Phone Dependence

According to Leung [46], the mobile phone addiction index scale was used to measure the mobile phone dependence in this study. The mobile phone addiction index scale was translated by Chinese scholars into Chinese, and later it was revised by psychology experts to form the final formal scale. The Chinese version contains 4 factors and 17 items, and each of the items is scored by 5-point Likert scale with 1 = never, 2 = rarely, 3 = occasionally, 4 = often, and 5 = always and the higher the score, the higher the degree of mobile phone dependence. In this study, the Cronbach’s alpha of the mobile phone addiction index scale was 0.89.

#### 2.3.3. Self-Esteem

The self-esteem scale developed by Rosenberg was used as the tool for measuring the level of self-esteem. The scale consists of 10 items; each item uses a 4-point scale from extremely inconsistent to extremely consistent and the higher the score, the higher level of self-esteem. Before calculating the total score, the scoring of items needed to be reversed. Rosenberg’s self-esteem scale has great reliability and validity [52,53,54,55]. In this study, the Cronbach’s alpha of Rosenberg’s self-esteem scale was 0.86.

#### 2.3.4. Social Support

The social support rating scale was developed in 1986 and has been used widely in China [45]. In this study, the social support rating scale was used to detect the degree of social support and the usage of support in social life. The social support rating scale consists of 10 items and 3 factors, namely objective support, subjective support, and support usage, separately. Objective support refers to the real and realistic support; subjective support means the self-perceived support, and support usage means the social support that is used when individuals seek social support and the higher the total score, the higher the degree of social support. In this study, the Cronbach’s alpha of the social support rating scale was 0.89.

### 2.4. Statistical Analysis

SPSS17.0 was used to analyze the data. Many scholars use SPSS to test moderated mediation model or mediated moderation model [4]. Firstly, a preliminary descriptive statistical of each variable was performed, including mean, standard deviation, and correlation coefficient. Secondly, the hypothetical moderated mediation model was tested.

## 3. Results

### 3.1. Common Method Biases Test

The problem of common method biases has been attracting more and more attention. The common rater, common measurement context, or the common item context may cause common method biases, and it is important to test the common method biases [56]. In this study, the data were derived from the self-rated questionnaire, so the results may be affected by the common method biases. In order to ensure the accuracy of the conclusions, it was necessary to test the common method biases before the formal analysis. The Harman’s single-factor test was used to load all the variables into an exploratory factor analysis [57]. The results of the nonrotating factor show that there were 19 factors with the eigenvalue greater than 1, and the variance of the first factor explained was 12.97%, which was far lower than the critical standard of 40%, indicating that there was no obvious common method bias.

### 3.2. Descriptive Statistics

The results of the descriptive statistics and correlation matrix are shown in Table 2.

According to Table 2, mobile phone dependence was negatively correlated with subjective well-being and self-esteem, and there was no significant correlation with social support. Subjective well-being was positively correlated with social support and self-esteem. There was a positive correlation between self-esteem and social support. The mediator requires a significant correlation with the independent variable and the dependent variable, but as a moderator, it is possible to correlate with the independent and dependent variables, but better not. According to that, self-esteem as a mediator for mediating analysis and social support as a moderator for moderating analysis are reasonable.

The overall situation about the subjective well-being of college students in China is shown in Table 3. Furthermore, the significant tests of age and gender are shown in Table 3. However, because race and socioeconomic status are not significant for college students in China, the results for them are not shown in Table 3.

According to Table 3, there were significant differences in gender and age. So, they needed to be controlled if the impact of other independent variables on subjective well-being needed to be explored.

### 3.3. Moderated Mediation Model Test

According to Table 3, the Bonferroni values of gender and age are significant. So, they needed to be controlled by the regression. Hierarchical regression was used to control gender and age to discuss the impact of other variables on subjective well-being. Here, only the influence results of independent variables with gender and age on dependent variables are listed. Here, demographic variables (i.e., gender and age) were taken as the first level, and other variables (including mobile phone dependence, social support, and self-esteem) were taken as the second level.

Many scholars believe that in the analysis of the moderating effect, the independent variable and moderator should be centered or standardized. Based on that, interactive items (social support × self-esteem) were further produced to facilitate the follow-up analysis. Therefore, in the next test of the moderated mediation model, the variables were a standardized *Z* score. According to prior studies [46,58,59], four specific steps were introduced as follows:

(1) To perform the hierarchical regression analysis of the dependent variable (subjective well-being) on the independent variable (mobile phone dependence) and the moderator (social support) with gender and age, the coefficient of mobile phone dependence was significant.

(2) To conduct the hierarchical regression analysis of the mediator (self-esteem) on the independent variable (mobile phone dependence) with gender and age, the coefficient of mobile phone dependence was significant.

(3) To perform the hierarchical regression analysis of the dependent variable (subjective well-being) on the independent variable (mobile phone dependence), the mediator (self-esteem), and the moderator (social support) with gender and age, the coefficient of self-esteem was significant.

(4) To conduct the hierarchical regression analysis of the dependent variable (subjective well-being) on the independent variable (mobile phone dependence), the moderator (social support), the mediator (self-esteem), and the mediator ×moderator (social support ×self-esteem) with gender and age, the coefficient of social support × self-esteem was significant.

Accordingly, the above four equations were tested in turn, and the specific results are shown in Table 4 and Table 5.

As shown in equation 1 of Table 4, mobile phone dependence had a negative effect on subjective well-being (β = −0.26, *t* = −6.39, *p* < 0.01), indicating that mobile phone dependence was an obstacle to subjective well-being. In equation 2, mobile phone dependence predicted self-esteem negatively (β = −0.11, *t* = −2.78, *p* < 0.01), which meant serious mobile phone dependence would reduce self-esteem.

As shown in equation 3 of Table 5, self-esteem positively predicted subjective well-being (β = 0.09, *t* = 2.03, *p* < 0.05), indicating that improvement in self-esteem would enhance the subjective well-being; in addition, the role of mobile phone dependence on subjective well-being was significant. According to the results of Table 5, self-esteem played a partial mediating effect between mobile phone dependence and subjective well-being; in other words, mobile phone dependence did not only directly affect subjective well-being, but also affected subjective well-being indirectly by mobile phone dependence.

As shown in equation 4 of Table 5, the interactive items (social support × self-esteem) had a significant effect on subjective well-being (β = 0.10, *t* = 2.50, *p* < 0.05), which showed that social support moderated the relationship between self-esteem and subjective well-being; concretely speaking, social support moderated the latter half of the mediation process (see Figure 2). Apart from that, the explanatory rate increased from 17% to 22% and Δ*R*^2^ = 0.05.

Although the results confirm the moderated mediation model, the moderating effect needed further analysis. In order to clarify the specific mechanisms of the moderator (social support), self-esteem was used as the abscissa; subjective well-being was used as the ordinate, and the *Z* scores of social support were equal to 1, 0, and −1 to make an interaction graph (see Figure 3). The slope in Figure 3 intuitively reflects the effect of self-esteem on subjective well-being.

Figure 3 shows that the effect of self-esteem on subjective well-being gradually enhanced with an increase in social support, which showed that mobile phone dependence affected subjective well-being indirectly by mobile phone dependence enhanced with a gradual increase in social support. When the *Z* score of social support = −1, the effect of self-esteem on subjective well-being was −0.03. When the *Z* score of social support = 0, the effect of self-esteem on subjective well-being was 0.07; that is, self-esteem increased 1 standard deviation, and subjective well-being increased by 0.07 standard deviations. When social support = 1, the effect of self-esteem on subjective well-being was 0.17; in other words, subjective well-being increased by 0.17 standard deviations, while self-esteem increased 1 standard deviation. In addition, at the same level of self-esteem, the higher the social support, the higher the subjective well-being, and with the increase in self-esteem, the feature was more obvious.

In summary, the moderated mediation model proposed in this study has been confirmed. In this model, self-esteem was the mediator between mobile phone dependence and subjective well-being; social support was the moderator, which moderated the second path of the mediation process; concretely speaking, the effect of mobile phone dependence affected subjective well-being indirectly by mobile phone dependence enhanced with a gradual increase in social support.

## 4. Discussions

### 4.1. Analysis of Mediating Effect of Self-Esteem

This study examined the mediating effect of self-esteem and found that self-esteem played a partial mediating effect between mobile phone dependence and subjective well-being. That is to say, mobile phone dependence not only had a direct influence on subjective well-being, but also influenced subjective well-being indirectly by self-esteem. In the process of the mediating effect, self-esteem played a linking role between mobile phone dependence and subjective well-being. On the one hand, it reflected the negative relationship between mobile phone dependence and self-esteem; on the other hand, it reflected the positive influence between self-esteem and subjective well-being. Either the former negative relationship or the latter positive relationship was consistent with previous studies [60,61,62].

The motive of mobile phone use was divided into habitual use and instrumental use [8,63]. The study authors thought that the former was a habit, a way of looking for partnership and killing time, and the latter was thought to search for information and the feeling of awakening [8,63]. The motive of habitual use was more likely to lead to mobile phone dependence than instrumental use [64]. Apart from that, the motivation of mobile phone use was divided into interpersonal motivation and entertainment motivation, and entertainment motivation has a positive correlation with mobile phone dependence [65]. According to the above view, those mobile-phone-dependent people were more likely to spend a lot of time on mobile entertainment. Therefore, their life, work, and studies will be negatively affected, which will inevitably decrease self-esteem and subjective well-being.

There were several advantages of investigating the mediating effect of self-esteem; on the one hand, it prompted the studies to find out which factors will influence subjective well-being and helped to understand the complex mechanisms of mobile phone dependence on subjective well-being. On the other hand, it provided a reference for us to improve the subjective well-being of college students. It was suggested to not only limit the excessive use of mobile phones, but to also improve the self-esteem of college students.

### 4.2. Analysis of the Moderation Effect of Social Support

In this study, we further discussed whether the mediating process will be affected by the moderator. Based on the original mediation model, a moderator (social support) was added to construct a relatively complex moderated mediation model, which explored the role of social support in the mediating effect. The results show that social support has a positive impact on subjective well-being; that is, social support will increase subjective well-being. In addition, the result shows that social support has a significant effect on the mediating effect; that is, social support moderated the mediating process. Social support moderated the second path of the mediation, and the impact of self-esteem on subjective well-being was enhanced with the increase in social support.

According to ecological systems theory, this study found that the individual’s subjective well-being was not only influenced by self-esteem, but also by social support. In a good social support system, the individual was more likely to obtain help and support, including material and spiritual support [66]. The help will promote the development of the individual, and the individual can feel warmth and enjoyment, which will improve the individual’s evaluation of self-worth; that is, the self-esteem of individuals with high social support will be high, and higher social support and self-esteem will make individuals reach a higher subjective well-being [67]. Therefore, by improving the subjective well-being of college students, various factors should be considered to greatly improve the individual’s subjective well-being. This also fully shows that people live in an ecological system, which is not only affected by individual variables, but also by environmental variables; the result is the interaction between the two, and neither of them is dispensable.

The discussion of the moderator of social support helped us to understand the mechanisms of mobile phone dependence on subjective well-being. In addition, the model shows how the mediating variable was moderated by the moderator. Under the condition of higher social support, self-esteem can significantly improve the subjective well-being of college students. Therefore, in raising the subjective well-being of college students, more attention should be paid to cultivating self-esteem. When social support was low, the impact of self-esteem on subjective well-being was relatively limited. College students should be helped to receive more social support. This also shows that some problems should be solved according to the specific conditions to improve their subjective well-being.

### 4.3. The Relationship among Mobile Phone Dependence, Self-Esteem, Social Support, and Subjective Well-Being

Before conducting the analysis of the moderated mediation model, descriptive statistics were made for all the variables. A correlation analysis showed that mobile phone dependence negatively affected self-esteem and subjective well-being. In addition, self-esteem positively predicted social support and subjective well-being. It indicated that the increase in mobile phone dependence reduced the individual’s positive self-evaluation and positive emotional experience, which was consistent with previous studies [37].

After confirming that mobile phone dependence negatively predicted subjective well-being, the mediator (self-esteem) and the moderator (social support) were incorporated to construct a moderated mediation model to further explore the mechanisms of mobile phone dependence on subjective well-being.

For the management of mobile phone dependence of college students, people needed to pay more attention to the personality characteristics of different students and the self-esteem of students. People should not blindly educate students, but increase social support and create a good atmosphere on campus and in society. Only in this way can they improve their subjective well-being.

### 4.4. Limitations

There are also some limitations that should be noted about this study. Firstly, the mechanisms of mobile phone dependence on subjective well-being are not comprehensive. In this study, only one mediator (self-esteem) and one moderator (social support) were added to perform the analysis. However, there are some variables, which were not included in this study, but still also affect the individual’s subjective well-being. Secondly, the self-report method is too subjective, which will restrict the validity of the data due to social desirability. So, further studies can adapt a more objective approach for collecting the data. Thirdly, only hierarchical regression was used to analyze the moderated mediation effects. In fact, the structural equation model is often used to examine the model holistically, which can be improved in the future. Finally, this is a cross-sectional study; whether these conclusions can be applied to longitudinal studies is not clear. Therefore, future studies may be longitudinal in nature to strictly confirm the causal relationships among these variables.

## 5. Conclusions

According to the ecological systems theory, this study explored the relationship between mobile phone dependence and subjective well-being. Self-esteem plays a partial mediating effect between mobile phone dependence and subjective well-being. Mobile phone dependence not only has a direct influence on subjective well-being, but also influences subjective well-being indirectly by self-esteem. The mediating effect of self-esteem between mobile phone dependence and subjective well-being is moderated by social support. Social support moderates the second path of the mediation and the higher the social support, the greater the degree of self-esteem on subjective well-being.

In the process of improving college students’ subjective well-being, more attention should be paid to cultivating college students’ self-esteem. However, when social support is low, self-esteem has a relatively limited impact on subjective well-being. At this time, the level of college students’ acceptance of social support should be considered to improve. In order to deal with college mobile phone dependence more effectively, people need to pay more attention to the personality characteristics of different students and give them more social support, instead of blindly implementing some negative punishment measures, such as criticism, scolding, and corporal punishment, which may help them better adapt to campus life.

## Figures and Tables

**Figure 1 healthcare-11-01388-f001:**
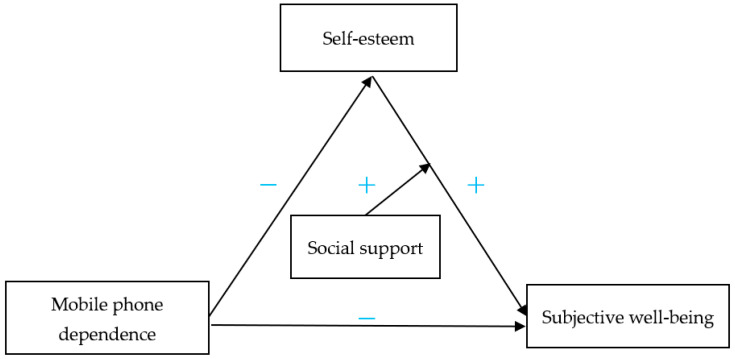
The framework of the study.

**Figure 2 healthcare-11-01388-f002:**
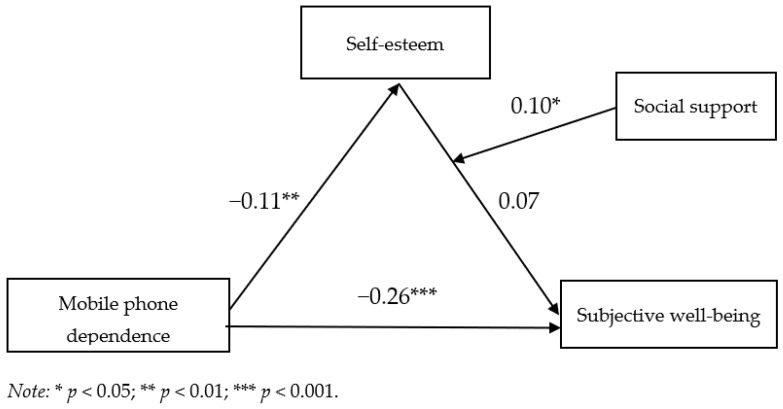
Relationship among mobile phone dependence, self-esteem, social support, and subjective well-being.

**Figure 3 healthcare-11-01388-f003:**
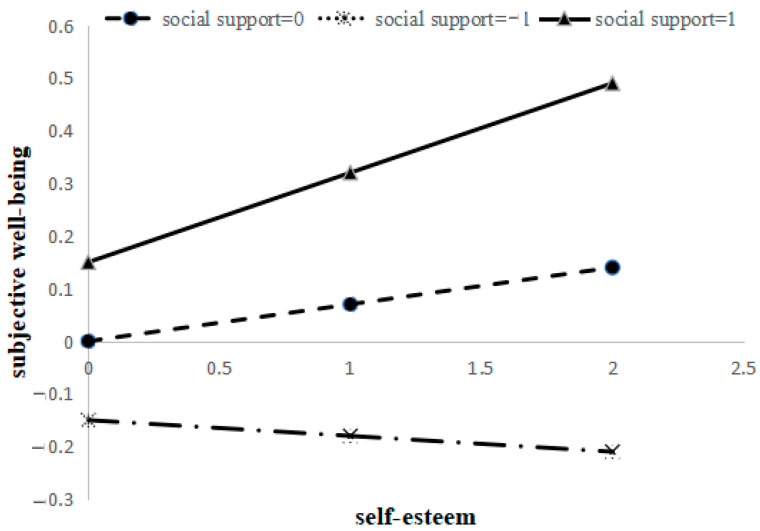
Analysis of moderating effect.

**Table 1 healthcare-11-01388-t001:** Descriptive statistics of demographic measurements.

Variable	Level	Number	Percentage
Gender	male	189	34.36%
	female	361	65.64%
Age	19	52	9.45%
	20	52	9.45%
	21	57	10.36%
	22	57	10.36%
	23	65	11.82%
	24	70	12.73%
	25	76	13.82%
	26	65	11.82%
	27	56	10.18%
Race	Han ethnicity	369	67.09%
	Non-Han ethnicity	181	32.91%
Socioeconomic status	Higher income	151	27.45%
	Middle income	203	36.91%
	Low income	196	35.64%

**Table 2 healthcare-11-01388-t002:** Descriptive statistics and correlation matrix.

Variables	*M*	*SD*	*Max*	*Min*	1	2	3	4
1 mobile phone dependence	38.78	10.56	85	17	1			
2 subjective well-being	71.46	8.33	100	18	−0.30 **	1		
3 social support	35.15	5.80	56	12	−0.06	0.18 **	1	
4 self-esteem	28.74	3.56	40	10	−0.14 **	0.18 **	0.40 **	1

*Note:* ** *p* < 0.01.

**Table 3 healthcare-11-01388-t003:** Comparison of demographic variables for subjective well-being.

		*M*	*SD*
Gender	*N* = 550	71.46	8.33
	male (*n* = 189)	73.21	6.82
	female (*n* = 361)	70.03	7.78
	*t*	3.45 **	
Age	19 (*n* = 52)	70.50	12.46
	20 (*n* = 52)	71.36	7.12
	21 (*n* = 57)	70.09	9.79
	22 (*n* = 57)	73.37	9.29
	23 (*n* = 65)	71.21	10.24
	24 (*n* = 70)	72.23	8.23
	25 (*n* = 76)	71.65	9.13
	26 (*n* = 65)	70.41	6.41
	27 (*n* = 56)	71.17	5.85
	*F*	2.57 *	

*Note:* * *p* < 0.05; ** *p* < 0.01.

**Table 4 healthcare-11-01388-t004:** Moderated mediation model test.

	Equation 1 (Dependent Variable: Subjective Well-Being)		Equation 2 (Dependent Variable: Self-Esteem)	
	β(t)	β(t)	β(t)	β(t)
Gender	0.16 (3.45 **)	0.14 (3.43 **)	0.01 (0.59)	0.26 (−0.50)
Age	0.17 (−2.57 *)	0.15 (−2.29)	0.06 (−1.09)	0.26 (−1.01)
Mobile phone dependence		−0.26 (−6.39 ***)		−0.11 (−2.78 **)
Social support		0.18 (4.29 ***)		
Self-esteem				
Social support × self-esteem				
*F*	2.86 *	25.76 ***	1.06	28.35 ***
*R^2^*	0.10	0.16	0.10	0.17
△*R*^2^	0.10	0.06	0.10	0.07

*Note:* * *p* < 0.05; ** *p* < 0.01; *** *p* < 0.001.

**Table 5 healthcare-11-01388-t005:** Moderated mediation model test.

	Equation 3 (Dependent Variable: Subjective Well-Being)		Equation 4 (Dependent Variable: Subjective Well-Being)	
	β(t)	β(t)	β(t)	β(t)
Gender	0.16 (3.45 **)	0.14 (3.43 **)	0.16 (3.45 **)	0.13 (3.24 **)
Age	0.17 (−2.57 *)	0.16 (−2.22 *)	0.17 (−2.57 *)	0.15 (−1.91)
Mobile phone dependence		−0.25 (−6.32 ***)		−0.26 (−6.40 **)
Social support		0.14 (3.35 **)		0.15 (3.47 ***)
Self-esteem		0.09 (2.03 *)		0.07 (1.72)
Social support × self-esteem				0.10 (2.50 *)
*F*	2.86 *	21.55 ***	2.86 *	19.17 ***
*R^2^*	0.10	0.17	0.10	0.22
△*R^2^*	0.10	0.07	0.10	0.12

*Note:* * *p* < 0.05; ** *p* < 0.01; *** *p* < 0.001.

## Data Availability

The datasets used and/or analyzed during the current study are available from the author on reasonable request.

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
