# Peer review of "The Relationship between Mobile Phone Dependence and Subjective Well-Being of College Students in China: A Moderated Mediation Model"

_healthcare, 2023, doi:10.3390/healthcare11101388_

Round 1
Reviewer 1 Report
After reading the revised version of the paper, I believe the manuscript has been significantly improved, especially in the method section. Only thing is missing. I strongly recommend the authors to explain the logics and reasons behind the research hypotheses, it needed to add at least one paragraph explanation for each hypothesis.
Why should mobile phone dependence improve subjective well-being? Your results show the negative impact of dependence on wellbeing!!!
Explain in 1 paragraph at least, why students’ self-esteem should hurt their subjective well-being? Your results and literature talking about the positive impact!!!!
Explain in 1 paragraph at least, why self-esteem plays a mediating role between cell phone dependence and subjective well-being? Explain. A minor change, be consistent with your terminology throughout of the paper, cell phone or mobile phone.
Explain in 1 paragraph at least, why social support plays a moderating role between self-esteem and subjective well-being?
Reviewer 2 Report
Congratulations to the authors for improving the submitted manuscript so much. For my part, I only consider small suggestions for improvement:
- Some variables involved in the study (gender, age, income) have no reflection in the theoretical framework.
- It is not explained why and for what the differentiation of Han nationality is measured.
- Table 1 could be replaced by a population pyramid, bar charts, and histograms.
